# Voluntary HIV Counselling and Testing Services: Knowledge, Attitudes, and Correlates of Utilisation among Young People in the Tema Metropolis, Ghana

**Emmanuel Anongeba Anaba [1,\*], Zita Buabeng [2] and Grace Adjei Okai [3]**

1   Department of Population, Family and Reproductive Health, School of Public Health, University of Ghana, Accra P.O. Box LG 13, Ghana
2   Nursing Department, 37 Military Hospital, Accra, Ghana
3   Department of Public Administration and Health Service Management, Business School, University of Ghana, Accra P.O. Box LG 78, Ghana
\*   Correspondence: emmaanaba24@gmail.com

**Abstract:** Voluntary HIV counselling and testing (VHCT) is a successful intervention for fostering early HIV detection, which is essential for the management of the disease. This study sought to determine the prevalence and factors that influence the utilization of VHCT services among young people. In this study, young people in the Tema Metropolis were cross-sectionally surveyed. The simple random sampling method was used to select the participants. The majority (60%) of the participants were unaware of VHCT, and 83% did not know of any VHCT centre. The majority (72%) of the participants indicated that VHCT was important, and 81% were willing to test for HIV if VHCT services were available in schools. Young people who knew that parental consent was not a requirement during VHCT were about two times more likely to have been tested for HIV (COR = 1.96; 95% CI: 1.05–3.63) compared to their counterparts. Additionally, young people who were willing to test in youth-friendly clinics (AOR = 2.87; 95% CI: 1.09–7.51) had higher odds of testing for HIV compared to their counterparts. The utilisation of VHCT services among young people in Tema was found to be very low. Additionally, young people's knowledge of VHCT services was below expectations.

**Keywords:** HIV/AIDS; utilisation; voluntary HIV counselling and testing; knowledge

## 1. Introduction

The proportion of young people (10–24 years) living with HIV is increasing globally. Young people are involved in risky sexual behaviours, including unprotected sex and multiple sex partners, which makes them vulnerable to HIV infection. Two out of every seven new HIV infections globally in 2019 were among young people [1]. In 2020, 410,000 new HIV infections were reported among young people [2]. The goal of eliminating HIV/AIDS worldwide by the year 2030 may not be achieved if young people are not prioritised. Young people should be the primary target of HIV transmission, susceptibility, impact, and behaviour change interventions [3]. Therefore, promoting access to HIV screening and treatment services among young people is crucial in the fight against the AIDS pandemic.

Voluntary HIV counselling and testing (VHCT) is an important intervention for HIV prevention as it may serve as an early entry point for prevention, cure and support for infected people. It is also crucial in promoting safe behaviour, providing personalized support and an opportunity for early detection [4]. Additionally, VHCT prevents delayed entry into HIV treatment, which ultimately leads to the desired reduction in deaths related to AIDS. Though VHCT is used as a diagnostic tool for asymptomatic AIDS, young people find it difficult to access it [5]. Evidence indicates that only 10% of young men and 15%

of young women know their HIV status [6]. Barriers to the utilization of VHCT services include lack of awareness, financial constraints, fear of testing positive and stigma. Other barriers include concerns about privacy, low-risk perceptions, and poor knowledge of HIV/AIDS [5,7,8].

In Ghana, HIV/AIDS remains a public health issue. For instance, 346,120 people in Ghana were HIV positive in 2020, and the prevalence among young people was 0.70% (42,016 people) [9]. In addition, 5211 new HIV infections were estimated to have occurred, accounting for more than a quarter of all new HIV cases in Ghana [9]. Adolescent HIV infections increased alarmingly by 500% in 2017 in the Tema Metropolis [10]. Thus, 143 cases of HIV infection were reported, including 60 males and 83 females. By 2025, the Ghana AIDS Commission wants to see an 85% decrease in the number of new HIV infections among young people, particularly among adolescent girls and young women. All sexually active young people in Ghana have unrestricted access to sexual and reproductive health services, including HIV testing.

However, there is a paucity of literature on HIV testing among young people in Ghana. The few existing studies focused on young people in rural areas [11–15], with little attention on those in urban areas, especially the Greater Accra Region, Ghana's capital city. Therefore, this study aimed to assess the prevalence and correlates of VHCT among young people in the Tema Metropolis, Ghana's industrial city. Stakeholders can leverage the findings of this study to inform HIV and AIDS policies and programmes. Additionally, the findings may inform the formulation of strategies that will help reduce the incidence of HIV among young people in Ghana.

## 2. Materials and Methods

### 2.1. Study Location and Design

We employed a descriptive cross-sectional study design. The Tema Metropolis is located in the south-eastern part of Ghana. The metropolis is the second largest populated district in the Greater Accra Region and the eleventh most populated area in Ghana. Four (4) sub-metropolitans are in the Tema Metropolis, including Tema West, East, South, and Central [16].

### 2.2. Population and Sampling

The target population for this study were young people (aged 10–24 years) residing in the Tema Metropolis. The Ghana Statistical Service (GSS) annual projections estimated that 67,861 young people resided in the Tema Metropolis, representing 20.3% of the total population in the metropolis. Version 3.01 of the OpenEpi Info software was employed to calculate the sample size for this study. The sample size was calculated using a design effect of one (1) and an assumed percentage of 91%, which is the response rate of a survey among young people in the area. The calculated sample size was 382. Ten (10) per cent of the calculated sample size ($n = 38$) was added to cater for non-response. The total sample size was 420 young people, including males and females. Study participants were selected using the simple random sampling method through the lottery method. The Tema Metropolis was purposely selected due to the high HIV prevalence among young people.

Theoretical Model

This study is guided by the Theory of Reasoned Action propounded by Martin Fishbein and Icek Ajzen in 1967. The theory aims to explain the relationship between attitudes and behaviours within human action. The theory postulates that an individual's decision to engage in a particular behaviour is influenced by attitudes, subjective norms and behavioural intentions. In this study, it was conceptualized that young people's uptake of VHCT services would be influenced by their attitudes towards HIV testing and perceptions about HIV testing as well as intentions to get tested. In addition, participant characteristics may also have a significant influence on utilization of HIV counselling and testing services. Therefore, the dependent variable in this study was tested for HIV. The independent variable includes attitudes towards VHCT, perceptions of VHCT and intentions to get tested for HIV, controlling for participant characteristics.

*2.3. Instruments*

A structured questionnaire was used to collect the data. The questionnaire was divided into four sections. The first section comprised the socio-demographic characteristics of the participants. Sex was coded as a dummy variable (1 = male and 0 = female). Religion was coded as 1 = Christian, 2 = Islamic, 3 = Traditional, and 4 = others. Age was a continuous variable. Ethnicity was coded as 1 = Ga, 2 = Asante, 3 = Ewe, and 4 = others. Residential status was coded as 1 = grandparent, 2 = father, 3 = mother, 4 = father only, 5 = mother only, 5 = brother or sister, and 6 = others. The second section of the questionnaire collected information on attitudes and perceptions of VHCT using six nominal items, all coded (1 = yes, 2 = no, and 3 = do not know). The final section of the questionnaire collected information on the uptake of VHCT services using a single item coded as (1 = yes, 2 = no). The questionnaire was reviewed and validated by experts in the subject area.

*2.4. Data Collection*

The questionnaire was administered to young people who met the inclusion criteria and consented. Where applicable, questions were verbally explained in the requisite local language for respondents who could not read and understand English. Data collection was performed by trained research assistants. On average, twenty minutes was spent completing the questionnaire. The questionnaire was administered in a private location.

*2.5. Ethical Considerations*

Before the survey, approval letters were sought from the Ghana Health Service Ethical Review Boards. Furthermore, participants' approval was obtained, and participation in this study was completely voluntary. The privacy and confidentiality of the participants were assured, and no identifiable information was collected. Completed questionnaires were kept under lock and key and were only accessible to the principal investigator. There was no form of compensation, either monetary or gifts for the participants.

*2.6. Statistical Analysis*

The data were analysed using Statistical Package for Social Sciences (SPSS) software, version 23. Retrieved questionnaires were cross-checked for errors and completeness, then coded and keyed into the software. Descriptive statistics were computed using frequencies and percentages. Binary logistic regression analysis was employed to assess the factors associated with the uptake of VHCT services. We used standard (Enter method) in selecting the independent variables into the binary logistic regression model.

## 3. Results

*3.1. Socio-Demographic Characteristics of Respondents*

It was found that the majority (69%) of the participants were females, and 59% of them were aged 14 to 17. The mean age was 17.3 years, with a standard deviation of 1.5. About three in ten participants were of the Ga ethnic group. Specifically, 91% of the participants professed Christianity, and 55% stayed with both parents (Table 1).

**Table 1.** Participants' characteristics.

| Characteristics (*n* = 360) | Frequency (*n*) | Percentage (%) |
|---|---|---|
| Sex | | |
| Male | 116 | 31 |
| Female | 259 | 69 |
| Age (years) | | |
| 14–17 | 215 | 59 |
| 18–19 | 127 | 35 |
| 20–24 | 21 | 6 |
| Ethnicity | | |
| Ga | 101 | 28 |
| Asante | 93 | 25 |
| Ewe | 82 | 22 |
| Others | 91 | 25 |
| Religion | | |
| Christianity | 338 | 91 |
| Islam | 34 | 9 |
| Live with | | |
| Grandparents | 34 | 9 |
| Both parents | 206 | 55 |
| Father alone | 21 | 6 |
| Mother alone | 73 | 20 |
| Siblings | 20 | 5 |
| Others | 20 | 5 |
| Have been tested for HIV | | |
| Yes | 48 | 13 |
| No | 312 | 87 |

*3.2. Young People's Awareness and Perceptions of VHCT Services*

It was found that the majority of the participants (60%) were unaware of VHCT services. Moreover, the majority (56%) of the participants were unaware that VHCT is optional. In addition, 52% of the participants knew that VHCT involves counselling and 60% knew that it is confidential. However, eight in ten participants were unaware of a VHCT centre, and 57% did not know that one could get tested for HIV without parental consent (Table 2).

*3.3. Young People's Attitudes toward VHCT Services*

The findings showed that the majority (70%) of the participants perceived VHCT to be important. Similarly, 72% of the participants felt that VHCT was necessary. Eight in ten participants indicated that they would get tested if VHCT services were available in school-based health clubs. Moreover, 81% of the participants indicated that they would get tested if HIV testing services were free of charge, while 64% of the respondents indicated that they would get tested if VHCT services were available in youth-friendly clinics (health facilities that provide age-appropriate and tailor-made health services to young people) (Table 3).

**Table 2.** Young people's awareness and perceptions of VHCT.

| Item | Frequency (*n*) | Percentage (%) |
|---|---|---|
| Heard about VHCT | | |
| No | 218 | 60 |
| Yes | 146 | 40 |
| VHCT is not compulsory | | |
| No | 212 | 56 |
| Yes | 158 | 44 |
| VHCT involves counselling | | |
| No | 171 | 48 |
| Yes | 185 | 52 |
| VHCT is confidential | | |
| No | 143 | 40 |
| Yes | 218 | 60 |
| Know VHCT centre | | |
| No | 304 | 83 |
| Yes | 62 | 17 |
| Parents' consent is not needed | | |
| No | 331 | 57 |
| Yes | 157 | 43 |

**Table 3.** Young people's attitudes toward VHCT services.

| Item | Frequency (*n*) | Percentage (%) |
|---|---|---|
| VHCT is important | | |
| No | 110 | 30 |
| Yes | 251 | 70 |
| VHCT is necessary | | |
| No | 81 | 28 |
| Yes | 258 | 72 |
| I will test if VHCT is in schools | | |
| No | 77 | 19 |
| Yes | 296 | 81 |
| I will test if VHCT is free | | |
| No | 68 | 19 |
| Yes | 297 | 81 |
| I will test if VHCT is in youth clinics | | |
| No | 131 | 36 |
| Yes | 231 | 64 |

*3.4. Predictors of Utilization of VHCT Services among Young People*

The multiple logistic regression analysis showed that two factors were found to be statistically significant predictors of the utilization of VHCT services among young people. Two regression models, including crude and adjusted models, were computed. In the crude model, young people who knew that parental consent was not a requirement for accessing VHCT services were about two times more likely to have been tested for HIV (COR = 1.96; 95% CI: 1.05–3.63) compared to their counterparts. In the adjusted model, young people who were willing to test in youth-friendly clinics (AOR = 2.87; 95% CI: 1.09–7.51) had higher odds of testing for HIV compared to their counterparts (Table 4).

**Table 4.** Factors associated with the uptake of VHCT services among young people in the Tema Metropolis.

| Covariate/Exposure | Crude Analysis OR (95% CI) | Adjusted Analysis OR (95% CI) |
|---|---|---|
| Sex | | |
| Male | 1 (ref) | 1 (ref) |
| Female | 0.63 (0.34–1.19) | 0.71 (0.31–1.60) |
| Age | | |
| | 1.09 (0.89–1.34) | 0.90 (0.69–1.17) |
| Ethnicity | | |
| Ga | 1 (ref) | 11 (ref) |
| Asante | 0.61 (0.26–1.44) | 0.44 (0.15–1.24) |
| Ewe | 0.78 (0.34–1.80) | 0.70 (0.25–1.93) |
| Others | 0.65 (0.28–1.53) | 0.87 (0.32–2.41) |
| Religion | | |
| Christianity | 1 (ref) | 11 (ref) |
| Muslim | 1.17 (0.67–4.52) | 1.11 (0.29–4.27) |
| Living status | | |
| grandparents | 1 (ref) | 11 (ref) |
| Both parents | 0.58 (0.21–1.57) | 0.43 (0.14–1.34) |
| Father only | 0.50 (0.09–2.75) | 0.63 (0.09–4.18) |
| Mother only | 0.58 (0.18–1.83) | 0.53 (0.14–1.97) |
| Brother/sister | 2.07 (0.55–7.70) | 3.03 (0.59–15.53) |
| Others | 0.90 (0.19–4.12) | 0.63 (0.11–3.40) |
| Heard about VHCT | | |
| No | 1 (ref) | 11 (ref) |
| Yes | 1.54 (0.83–2.84) | 1.66 (0.72–3.82) |
| VHCT is not compulsory | | |
| No | 1 (ref) | 11 (ref) |
| Yes | 1.23 (0.66–2.28) | 1.19 (0.46–3.05) |
| VHCT is important | | |
| No | 1 (ref) | 11 (ref) |
| Yes | 1.14 (0.57–2.26) | 1.21 (0.40–3.69) |
| VHCT involves counselling | | |
| No | 1 (ref) | 11 (ref) |
| Yes | 0.87 (0.46–1.62) | 0.69 (0.26–1.80) |
| VHCT is confidential | | |
| No | 1 (ref) | 11 (ref) |
| Yes | 0.83 (0.44–1.56) | 0.85 (0.302–2.43) |
| Know VHCT centre | | |
| No | 1 (ref) | 11 (ref) |
| Yes | 1.51(0.72–3.17) | 1.52 (0.56–4.14) |
| VHCT is necessary | | |
| No | 1 (ref) | 11 (ref) |
| Yes | 1.03 (0.51–0.10) | 0.71 (0.26–1.88) |
| Parental consent is not needed | | |
| No | 1 (ref) | 11 (ref) |
| Yes | 1.96(1.05–3.63) * | 1.28 (0.59–2.78) |
| I will test if VHCT is in schools | | |
| No | 1 (ref) | 11 (ref) |
| Yes | 0.96(0.44–2.11) | 0.52 (0.59–2.78) |
| I will test if VHCT is free | | |
| No | 1 (ref) | 11 (ref) |
| Yes | 1.15(0.51–2.59) | 1.17 (0.31–4.38) |
| I will test if VHCT is in the adolescent clinic | | |
| No | 1 (ref) | 1 (ref) |
| Yes | 1.72 (0.85–3.45) | 2.87 (1.09–7.51) * |

* $p < 0.05$.

## 4. Discussion

This study sought to assess young people's knowledge, attitudes, and utilization of VHCT services. The results showed that young people's knowledge of VHCT was low. In particular, most young people were unaware of VHCT as well as had poor knowledge of VHCT outlets. This finding has been confirmed by a previous study in Accra [17]. On the other hand, these findings are contrary to prior findings in other sub-Saharan African countries. For example, in Nigeria, [18] found that the majority of the youth had adequate knowledge about VHCT services, with many indicating the mass media as their main source of information. In Ethiopia, it was also found that many adolescents had adequate knowledge of VHCT [19]. The differences in findings can be attributed to differences in contextual factors. A possible reason for the low awareness of VHCT among young people in Ghana may include low public education about testing. The low awareness of VHCT may explain why the utilization of VHCT services among young people was low. This implies that many young people, including those who are infected, may not know their HIV status. Studies have shown that knowledge of HIV testing is a predictor of the uptake of HIV testing services [7,19].

Generally, the majority of the participants demonstrated positive attitudes towards VHCT. For example, more than half of the respondents were willing to test for HIV if VHCT services were free of charge, available in schools or youth clinics. A higher proportion of the youth preferred school-based testing centres to testing in youth-friendly clinics. A plausible reason is that youth-friendly clinics in Ghana are located within the premises of health facilities, which may discourage young people from access VHCT services due to stigma. There is evidence to show that the location of youth-friendly clinics engender inconvenience for Ghanaian young people to access adolescent-health services.

Although the majority of the respondents had positive attitudes towards VHCT, their attitudes did not translate into the uptake of VHCT services. Thus, less than half of the respondents had been tested for HIV, hence they knew their status. This finding is supported by existing studies [20,21]. For instance, Gadegbeku (2013) found that only 6% of adolescents in Accra had ever visited the VHCT centre, out of which 4% went there for HIV testing, while the remaining 2% paid a visit to their friends working at the centre [17]. Similarly, Ibrahim et al. (2013) found that only one in ten young people in Nigeria had been tested for HIV [22].

On the contrary, a similar study in South Africa found that the majority of adolescents had been tested for HIV [23]. The low uptake of VHCT services as reported in this study may be attributed to other factors than just low knowledge. For instance, the prevalence of HIV in a particular country may influence the uptake of testing services. South Africa, for example, is one of the countries in Africa with a high HIV/AIDS prevalence, hence, it is not surprising to find that the majority of adolescents have been tested. High prevalence implies that more effort and resources would have been invested in promoting early detection through testing. Moreover, more public education and awareness creation strategies would have been implemented to promote the uptake of testing services.

In addition, the findings showed that socio-demographic factors were not associated with the utilization of VHCT services [14]. These findings are contrary to existing findings [8,24]. A similar study in Nigeria found an association between urban residence, secondary education, and the uptake of HIV testing services among young people [22]. Gender and age were also found as significant predictors of HIV testing in South Africa [23]. A possible reason for the differences in the findings is contextual disparities.

### 4.1. Implications of the Findings

Stakeholders must give these findings the attention they require. The Ghana Health Service can collaborate with the Ghana AIDS Commission and Ghana Education Service to intensify HIV testing and education among young people. For example, VHCT services should be brought closer to young people through outreach activities in schools and communities. An emphasis should be placed on helping young people to understand that all people are susceptible to HIV and that testing is the only way to know their HIV status. Additionally, young people should be encouraged and motivated to utilize HIV testing services by educating them on the protocols involved and assuring them of privacy and confidentiality. These efforts should be complemented by educating young people about HIV treatment options, including antiretroviral therapy, coupled with allaying their fears and misconceptions associated with HIV testing. These efforts would help more young people to take up HIV testing services, and those who test positive should be put on medication. These initiatives would help Ghana to achieve the 95–95–95 targets by 2030 [25] as well as reduce new HIV infections among young people by 85% by 2025 [9].

### 4.2. Strengths and Limitations

The findings of this study make a modest contribution to the literature on HIV testing in Ghana. Stakeholders can leverage the information to inform HIV policies and programmes as well as future research. Although this study provides relevant information for practice, policy, and research, it is not devoid of limitations. The Tema Metropolis is an urban area with many industrial activities. The location has unique characteristics; hence, the findings cannot be generalized to other settings. Additionally, association does not mean causation, so the findings should be interpreted with caution.

### 5. Conclusions

This study has demonstrated that there was low knowledge and utilization of VHCT among young people in the Tema Metropolis. However, young people demonstrated positive attitudes towards VHCT services. The uptake of VHCT was not associated with socio-demographic factors. The findings of the study suggest that stakeholders would have to develop strategies to help increase the uptake of VHCT services among vulnerable populations, such as young people. Promoting the uptake of HIV testing services among young people would be crucial in the fight against the HIV/AIDS epidemic and contribute to achieving the Sustainable Development Goal three, which among others seeks to end HIV/AIDS by the year 2030. Therefore, the findings of this study have implications for HIV policies, programming, and research. This study employed a quantitative approach which is unable to explore the lived experiences of young people regarding HIV testing. Therefore, future studies should investigate the barriers and facilitators of utilisation of HIV testing services among young people in Ghana.

**Author Contributions:** Conceptualization, Z.B. and E.A.A.; methodology, E.A.A.; software, E.A.A.; validation, Z.B., E.A.A. and G.A.O.; formal analysis, E.A.A.; investigation, Z.B.; resources, Z.B.; data curation, Z.B.; writing—original draft preparation, E.A.A.; writing—review and editing, Z.B. and G.A.O.; visualization, G.A.O.; supervision, E.A.A.; project administration, Z.B.; funding acquisition, Z.B. All authors have read and agreed to the published version of the manuscript.

**Funding:** This research received no external funding.

**Institutional Review Board Statement:** All subjects gave their informed consent for inclusion before they participated in the study. The study was conducted in accordance with the Declaration of Helsinki, and the protocol was approved by the Ethics Committee of Ghana Health Service (GHS-ERC: 003/09/20).

**Informed Consent Statement:** Informed consent was obtained from all subjects involved in the study. Written informed consent has been obtained from the patient(s) to publish this paper.

**Data Availability Statement:** The data presented in this study are available on request from the corresponding author. The data are not publicly available due to privacy.

**Acknowledgments:** The authors would like to thank all the participants of this study.

**Conflicts of Interest:** The authors declare no conflict of interest.

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
