# Peer review of "Voluntary HIV Counselling and Testing Services: Knowledge, Attitudes, and Correlates of Utilisation among Young People in the Tema Metropolis, Ghana"

_2673-995X, doi:10.3390/youth2040034_

Round 1
Reviewer 1 Report
This paper is an interesting approach to HIV Voluntary Counseling and Testing (VHCT) to encourage early detection of HIV, which is essential for disease management. The paper is comprehensive and grounded in the standards of the international scientific community. The bibliography is adequate and up to date. I believe it is of sufficient quality for publication. I would simply like to ask the authors to expand a little more on the conclusions.
Author Response
The authors would like to thank the reviewer for his/her commendation. The conclusion section of the manuscript has been expanded.

Reviewer 2 Report
Comment
This study sought to determine the prevalence of, and factors influencing, utilizations of VHCT services among youth.
A structured questionnaire was obtained to collect the data. However, it is not clear whether the questionnaire is statistically validated or at least validated by experts. In general, if the questionnaire proposes to measure the effect of an intervention, it is necessary to establish dimensions and standardize the results.
With respect to Multiple Logistic Regression, it would be advisable to have a theoretical model, establish which variable is dependent (dichotomous variable (0-1) and the independent variables or factors that influence the adoption of the service or not
This to group factors that could be more associated. Table 4 shows many factors and some associated, but there is no real model that answers the research question or the hypothesis.
It is recommended to describe what is meant by Youth Friendly Services. They are generally instances where young people feel more identified and a negotiation for the adoption of preventive measures is possible. However, the results establish that young people would prefer VHCT centers to be in their schools. Therefore, it could be understood that the friendly centers are failing to capture young people and on the other hand, VHCT are not recognized in most cases. Therefore, this requires additional analysis, or it might be the subject of another study.
Author Response
Reviewer 2 |
|
A structured questionnaire was obtained to collect the data. However, it is not clear whether the questionnaire is statistically validated or at least validated by experts. In general, if the questionnaire proposes to measure the effect of an intervention, it is necessary to establish dimensions and standardize the results |
The questionnaire was validated by experts. The questionnaire not measure an intervention. |
With respect to Multiple Logistic Regression, it would be advisable to have a theoretical model, establish which variable is dependent (dichotomous variable (0-1) and the independent variables or factors that influence the adoption of the service or not |
This study was not underpinned by any theoretical model. The dependent and independent variables have been stated in the methods section |
It is recommended to describe what is meant by Youth Friendly Services. They are generally instances where young people feel more identified and a negotiation for the adoption of preventive measures is possible. However, the results establish that young people would prefer VHCT centers to be in their schools. Therefore, it could be understood that the friendly centers are failing to capture young people and on the other hand, VHCT are not recognized in most cases. Therefore, this requires additional analysis, or it might be the subject of another study. |
Youth-friendly services have been explained in the discussion section. More information has been provided to explain why young people preferred school-based testing services to youth-friendly centres. |

Round 2
Reviewer 2 Report
Comment
I believe that the comments were answered. However, there is still a question regarding the methodological issue. Although the results are aligned with the implemented methodology. But a question remains as to how they conceptualized the regression model they built and presented in this manuscript:
Regarding the comment:
“With respect to Multiple Logistic Regression, it would be advisable to have a theoretical model, establish which variable is dependent (dichotomous variable (0-1) and the independent variables or factors that influence the adoption of the service or not”.
The researchers responded: This study was not underpinned by any theoretical model. The dependent and independent variables have been stated in the methods section
But the concern is that, if the researchers applied a Logistic Regression model, they had to propose a model that would theoretically associate the different independent variables with the dependent one.
On the other hand, logistic regression models allow applying methods for the selection of variables step by step (stepwise).
Stepwise regression can be achieved by testing one independent variable at a time and including it in the regression model if it is statistically significant. All potential independent variables can be included in the model and those that are not statistically significant can be eliminated. Some use a combination of both methods, and therefore there are several approaches to stepwise regression:
Forward selection: Starts with no variables in the model, tests each variable as it is added to the model, then keeps the ones considered most statistically significant, repeating the process until the results are optimal.
Backward elimination starts with a set of independent variables, eliminating one at a time, then testing to see if the eliminated variable is statistically significant.
The comment is to the effect that the researchers indicate that they presented a crude model and an adjusted one, which means that they selected variables and then adjusted. The final model presented by you in the manuscript would be expected to go through this process selecting the most plausible model.
Therefore, I think that for the solidity of your study you should mention how you selected the variables of the study and the criteria to propose the final model that you present.
Author Response
Thank you for this comment.
Please the authors have proposed the Theory of Reasoned Action as the theoretical underpinning for this study. Details have been provided in the methods section.
Thanks for the comment regarding selecting the variables in the regression model.
The authors used the (ENTER METHOD) of selection. We did not do a stepwise regression. This information has been included in the methods section.
Thanks
